# Detecting Progression of Melanocytic Choroidal Tumors by Sequential Imaging: Is Ultrasonography Necessary?

**DOI:** 10.3390/cancers12071856

**Published:** 2020-07-10

**Authors:** Kelsey A. Roelofs, Roderick O’Day, Lamis Al Harby, Gordon Hay, Amit K. Arora, Victoria M. L. Cohen, Mandeep S. Sagoo, Bertil E. Damato

**Affiliations:** 1Ocular Oncology Service, Moorfields Eye Hospital, London EC1V 2PD, UK; roelofs@ualberta.ca (K.A.R.); roderick.oday@gmail.com (R.O.); lamis.alharby1@nhs.net (L.A.H.); gordon.hay3@nhs.net (G.H.); amit.arora@nhs.net (A.K.A.); victoria.cohen@nhs.net (V.M.L.C.); mandeep.sagoo1@nhs.net (M.S.S.); 2Department of Ophthalmology, Ocular Oncology Clinic, Royal Victorian Eye and Ear Hospital, Melbourne 3002, Australia; 3NIHR Biomedical Research Centre for Ophthalmology at Moorfields Eye Hospital and University College London Institute of Ophthalmology, London EC1V 9EL, UK; 4Nuffield Laboratory of Ophthalmology, University of Oxford, London OX3 9DU, UK

**Keywords:** choroidal nevus, choroidal melanoma, ultrasound, optical coherence tomography, autofluorescence, tele-ophthalmology

## Abstract

*Purpose:* To determine if ultrasonography is necessary to detect progression of choroidal melanocytic tumors undergoing sequential multi-modal imaging with color photography, autofluorescence (AF) and optical coherence tomography (OCT). *Methods:* All patients with choroidal melanoma undergoing treatment at Moorfields Eye Hospital between January 2016 and March 2020 were reviewed to identify those with treatment deferred by ≥2 months. Tumors that showed progression prior to treatment, defined as an increase in (a) basal dimensions (b) thickness (c) orange pigment and/or (d) sub-retinal fluid, were included. Mushroom shape, Orange pigment, Large size, Enlargement and Sub-retinal fluid (MOLES) scores were assigned to all tumors at earliest date and date of treatment. *Results:* A total of 99 patients with a mean age of 66 years (range: 26–90) were included. The initial MOLES score was 1 in 2 cases, 2 in 23 cases, and ≥3 in 74 cases. Progression was detected with sequential color photography alone in 100% of MOLES 1/2 and 97% of lesions with a MOLES score of ≥3. When findings on AF and OCT were included, sensitivity for detecting subtle change without ultrasonography improved to 100% for MOLES 3 and 97% for MOLES 4/5. Only one patient included in this study had an isolated increase in thickness that may have been missed had sequential ultrasonography not been performed. Overall, the sensitivity for detecting progression with color photographs alone was 97% (95% CI 93–100%) and increased to 99% (95% CI 97–100%) by including autofluorescence and OCT. *Conclusions:* Monitoring of choroidal nevi, particularly those classified as MOLES 1 or 2 (i.e., low-risk or high-risk naevi), can be accomplished safely without the need for ultrasonography. The findings of this study may remove barriers to the implementation of tele-oncology clinics for the monitoring of choroidal melanocytic tumors.

## 1. Introduction

The last decade has seen advances in multimodal ocular imaging and in telemedicine platforms, allowing innovative delivery of care in several ophthalmic sub-specialties [1,2,3,4,5], including oncology [6,7,8]. This technology enhances opportunities for the remote monitoring of melanocytic choroidal tumors of uncertain malignancy. Such ‘virtual’ surveillance is especially useful for patients who live far from ocular oncology centers and may be adopted more widely as a result of the COVID-19 pandemic and its aftermath [9].

Choroidal nevi are common, with a prevalence ranging from 0.3% to 6.5% [10,11,12,13,14,15]. It is estimated that malignant transformation occurs in less than 1 in 8000 choroidal nevi/year [16], correlating to an adjusted lifetime risk estimate of 0.2% [17]. There is mounting evidence that some choroidal melanomas metastasize early [18,19]. Hence, the monitoring of choroidal nevi to identify early transformation to melanoma is widely recommended.

Over the last two decades, tremendous advances in both the resolution and speed of optical coherence tomography (OCT) imaging, coupled with its non-invasive nature, have led to its widespread adoption [20,21]. Similarly, wide-field imaging modalities, including fundus autofluorescence, are being used increasingly to aid in the identification of lipofuscin [22]. When used in combination with various other multi-modal imaging techniques, a great deal of information can be gathered. In the case of choroidal nevi, monitoring protocols often take advantage of this synergistic multi-modal approach by incorporating fundus photography, autofluorescence (AF), OCT and ultrasonography (US).

The intensity of monitoring is often stratified according to the number of clinical risk factors present [23,24,25,26]. Shields et al. recently updated their mnemonic of risk factors for the growth of choroidal nevi to melanoma using multimodal imaging to ‘To Find Small Ocular Melanoma Doing IMaging’ [27]. This mnemonic refers to thickness > 2 mm, fluid under the retina, symptoms, orange pigment, melanoma hollow on ultrasonography, and diameter > 5 mm; however, in many instances access to US remains limited in the community. Unlike fundus photography, AF and OCT, ultrasonography requires a highly skilled operator and, as such, many tele-oncology programs still require patients to travel to a central location where the imaging equipment and expertise are available [6]. The incentive to identify early progression of choroidal melanocytic tumors while minimizing unnecessary patient travel to larger centers has perhaps never been as great as it is in the current COVID-19 era.

The MOLES acronym, score and management protocol have recently been developed by the senior author (BD) to enable non-subspecialty assessment of melanocytic choroidal tumors without ultrasonography and other specialized equipment. The MOLES acronym represents Mushroom shape, Orange pigment, Large size, Enlarging tumor and Subretinal Fluid. Each of these is scored as 0, 1 or 2, and tumors are categorized according to the sum of these scores as: 0 ‘common nevi’, 1 ‘low-risk nevi’, 2 ‘high-risk nevi’ and >2 ‘probable melanomas’. Urgent referral to an ocular oncologist is advised only for patients with probable melanoma. We have recently validated the MOLES scoring system in a dataset of predominantly choroidal nevi [28] and have found it to be a highly sensitive tool in a large cohort of choroidal melanomas [29]. The aim of this study was to determine if ultrasonography is necessary to detect progression of choroidal melanocytic tumors undergoing sequential color photography, AF and OCT.

## 2. Methods

We reviewed all cases of choroidal melanoma treated at Moorfields Eye Hospital between January 2016 and March 2020 to identify those with prior sequential imaging spanning a time interval ≥2 months. Patients were excluded if the entire tumor margin was not visible on wide-field fundus photography (Optos California (Optos plc, Dunfermline, Scotland)). In patients with more than one melanocytic choroidal tumor, only the lesion that eventually progressed to treatment was considered.

Ocular imaging consisted of fundus photography and AF (Optos California), OCT (Heidelberg Spectralis (Heidelberg Engineering GmbH, Heidelberg, Germany)) and US (Acuson Sequoia (Siemens, Healthineers, Frimley, Camberly, UK)). With respect to device specifics, the Optomap^®^ pseudo-colour image was produced by combining the green (532 nm) and red (635 nm) channels, whereas the autofluorescence image was generated using only the green. The Heidelberg Spectralis OCT used a wavelength of 870 nm and the US was performed using a 15L8 transducer probe. These techniques complement one another, as they vary in imaging area, scanning depth and invasiveness. For instance, wide-field fundus photography and AF capture 200° of the fundus but are unable to provide detailed topographic information. Heidelberg OCT images a 30° field, but has the ability to provide a vertical resolution far exceeding that of any other imaging modality. Finally, US has the greatest vertical scanning depth and is particularly useful for imaging lesions that are too thick for OCT (>2–3 mm); however, it is also the most invasive of the imaging modalities.

Patient files were then reviewed for demographic data, including sex, tumor laterality, age, reason for delaying ocular therapy, and type of treatment. Tumors were categorized using MOLES system (Table 1 and Table 2). Scoring of the earliest images available and last images preceding treatment was performed by either KR or RO, both of whom have sub-specialty training in ocular oncology. Lipofuscin was given a score of 1 if detected by FAF alone and a score of 2 if visible on color photographs. Similarly, subretinal fluid was scored as 1 if detected only by OCT and as 2 if apparent in fundus photographs.

Longitudinal and transverse basal tumor dimensions were measured using the Optos calipers, converting pixel number to distance in millimeters using the horizontal disc diameter as a scale, assuming this to be 1.5 mm. Tumor thickness was assessed by B-scan ultrasonography, measuring the distance from the internal scleral surface to the thickest part of the tumor, excluding the retina.

Tumor growth was assessed by reviewing sequential photographs and comparing any change in distances between tumor margins and fundus landmarks, such as retinal blood vessels and the optic disc. The detection of tumor progression was categorized as being achieved by: (A) photography, if there was a visible increase in basal tumor dimensions and/or if confluent clumps of orange pigment became apparent (O = 2); (B) OCT, if traces of subretinal fluid were detected only with this imaging technique (S = 1); (C) fundus autofluorescence imaging, if hyperfluorescent dusting of lipofuscin appeared (O = 1); and (D) ultrasonography, if the measured tumor thickness increased by more than 0.5 mm.

The largest basal tumor diameter was estimated from longitudinal and transverse measurements, whichever was greater. In some cases, photographic growth was documented without any increase in largest basal dimension (LBD); this is because such growth did not always occur in the same direction as the LBD. Statistical analysis was performed using commercially available software (Stata Statisical Software. StataCorp LP, College Station, TX, USA). Sensitivity for various combinations of imaging modalities is reported with 95% confidence intervals (CI). This study was approved by the Moorfields Eye Hospital clinical audit department (No; 452) and was conducted in accordance with the declaration of Helsinki.

## 3. Results

The cohort consisted of 99 patients (46 females and 53 males) with a mean age was 66 years (range, 26–90). The tumor was located in the left eye in 52 cases (52.5%). At the date of earliest imaging, 74/99 (74.8%) tumors were <2.0 mm in thickness. The largest basal diameter was >6.0 mm in 69/99 (69.7%) tumors, leading to a MOLES ‘L’ score of 2 in 71/99 (71.7%) cases. The initial diagnosis was suspicious choroidal nevus, indeterminate lesion, or giant choroidal nevus in 82/99 (82.8%) cases (Table 3).

The initial MOLES score was 1 (2 cases) or 2 (23 cases). In these 25 probable nevi, an increase in the basal dimensions noted on photography was seen in all but one patient (24/25; 96.0%). In the case where there was no increase in basal dimensions (case 25), the development of clumps of orange pigment (O = 2), visible both on autofluorescence and fundus photography (1/25; 4.0%) was seen. Therefore, progression was detected in 100% of the initial MOLES 1/2 lesions without the use of US, AF or OCT (Figure 1).

A total of 36 cases had an initial MOLES score of 3 (36/99; 36.4%). In these cases, tumor progression was detected on color photography alone by (a) lateral tumor extension in 33/36 (91.7%) or (b) development of clumps of orange pigment (O = 2) in 2/36 (5.6%). Therefore, progression would have been detected using color photography alone in 35/36 cases (97.2%; 95% CI: 92–100%). In the final remaining case (case 61), progression was demonstrated by the appearance of trace sub-retinal fluid (S = 1), visible on OCT, together with an increase in thickness on US of 0.5 mm. As OCT allowed the detection of tumor progression in the only case without lateral extension on color photography, progression would have been detected without ultrasonography in all 36 (100%) tumors (Figure 2).

There were 38 lesions categorized as MOLES 4 or 5 (Figure 3). Three lesions did not show any signs of progression (cases 83, 84 and 99). In cases 83 and 99, choroidal melanoma was diagnosed at the first visit but the patients initially refused treatment and, after a delay of 0.4 and 0.2 years, respectively, they agreed to undergo plaque brachytherapy. Case 84 underwent a work-up to rule out a systemic primary and following negative investigations, was treated with plaque brachytherapy after a delay of 0.4 years.

In the remaining 35 tumors, progression occurred and was detected on color photography by (a) lateral extension in 30/35 (85.7%) or (b) an increase in orange pigment to O = 2 in 3/35 (8.6%). Therefore, color photography alone would have identified progression in 33/35 cases (94.3%; 95% CI: 87–100%). In one of the two remaining cases, progression was detected by the development of trace sub-retinal fluid detected on OCT in 1/35 (2.9%). Therefore, using a combination of color photography, AF and OCT, progression was detected without ultrasonography in 34/35 cases (97.1%; 95% CI: 92–100%). Only 1/35 (2.9%) tumor in this group (Case 81) demonstrated an increase in thickness (i.e., 0.7 mm), which was detected on US without a visible change noted on color photography or OCT (Figure 3).

## 4. Discussion

### 4.1. Main Finding

In this study, we found that progression of melanocytic choroidal tumors could be detected by sequential colour photography alone in 100% of lesions with a MOLES score of 1 or 2 (i.e., low-risk and high-risk naevi, respectively) and 97% of lesions with a MOLES score of ≥3. Failure to detect progression without ultrasonography would have occurred in only 1/99 (1%) case; this tumor had a MOLES score of 4, so according to the MOLES protocol, the patient would already have been under the care of an ocular oncologist, who would have had ready access to ultrasonography (Figure 4).

### 4.2. How Important Is Monitoring Tumor Thickness?

Tumor thickness is a well-documented indicator of malignancy in melanocytic choroidal tumors [27] and is also a predictor of metastatic death after treatment of choroidal melanoma [30]. Moreover, an increase in tumor thickness and/or basal diameter of a choroidal melanocytic lesion has historically been regarded as a reliable indicator of malignancy [31,32]. Thinner tumors (1.1–2.0 mm) show proportionally larger increases in basal diameter than in thickness (2.7 mm versus 1.0 mm, respectively) compared to thicker tumors (>3.0 mm), in which basal diameter and thickness increase with a 1:1 ratio (i.e., 1.0 mm versus 0.9 mm, respectively) [27]. Our results are in keeping with this finding, as they show that an increase in basal dimensions occurred in 96% of MOLES 1 and 2 lesions, which had a mean initial thickness of 1.6 mm. In this study, 36 cases with tumor progression did not show a corresponding increase in thickness on US, compared to only one case that grew on US but was missed by photography/AF/OCT.

In an effort the minimize reliance on ultrasonography, the MOLES scoring system categorizes lesions by size based on a combination of largest basal dimension and/or thickness (Table 1 and Table 2), which begs the question: how often do choroidal melanocytic tumors exhibit isolated growth in thickness without accompanying increases in basal dimensions or other signs of progression? Our study suggests that progression indicated by ultrasonography alone is rare, occurring in only in about 1% of cases. As our study was conducted at a tertiary, ocular oncology center, lesions were larger and had more indicators of malignancy than would be encountered in a typical community setting. We therefore had a reasonably large proportion of patients with MOLES scores ≥ 3 for whom referral to an ocular oncology center is recommended by the MOLES protocol. As tumors monitored in the community would be expected to be smaller, cases demonstrating isolated progression detected only on ultrasonography are likely to be even less common than the 1% found in this study.

In this study, photographic demonstration of growth in basal dimensions was determined by comparing the lesions’ extent in relation to vascular landmarks and the optic disc, as is usually done in clinical practice. Caliper placement for quantification of basal dimensions was challenging in some lesions, particularly those with indistinct margins. This introduced an undeniable level of variability and error in our LBD measurements and determination of change in LBD over time. Moreover, as an increase in lesion dimensions did not always occur along the meridian of the LBD, in isolation these values do not accurately represent the true degree of change in the lesion size. Therefore, it is important to emphasize that comparison of lesion margins in relationship to retinal vascular landmarks, rather than sequential measurement of LBD, should be used to identify growth on color photography.

### 4.3. Value of Ultrasonographic Evaluation in Addition to Measurement of Tumor Thickness

Low internal acoustic reflectivity (i.e., ‘ultrasonographic hollowness’) can aid the differentiation of choroidal nevi from melanomas [25,33,34] and is helpful in a sub-specialty ocular oncology practice when weighing the risks and benefits of treatment versus observation. With respect to echographic changes over time, Doro et al. found that nevi < 1.5 mm thick showed no change, whereas a progressive increase in hollowness on B-scan occurred in 18% of tumors with thickness > 1.5 mm [35].

Monitoring relatively small tumors with ultrasonography has several limitations. First, this examination may fail to detect thin tumors [8]. Second, with thin tumors, it frequently over-estimates lesion thickness [36]. Third, thickness measurements can be overestimated by the inadvertent inclusion of the retinal and/or scleral thickness or unintentional oblique scans. Fourth, the well-documented 0.5 mm inter- and intra-observer variability in tumor thickness measurement represents a large percentage change in the thickness of small tumors. This problem is illustrated by our case 62; this tumor grew from a dome to a mushroom shape, which suggests that the 0.2 mm increase in thickness was an under-estimate.

### 4.4. Optimal Method for Measuring Tumor Thickness

With small, posterior tumors, enhanced depth imaging OCT (EDI-OCT) may be a reasonable alternative in community practices. Enhanced depth imaging OCT can detect posterior scleral bowing in 5–14% of choroidal nevi [37,38], especially those with less/mixed pigmentation, posterior location, or a surrounding halo [37]. In such cases, ultrasonographic measurement may underestimate true tumor thickness although, in most instances, ultrasound overestimates the thickness of minimally elevated choroidal tumors. Our unpublished observations show that OCT is often more useful for measuring choroidal nevi <1 mm in thickness. However, measurement of tumor thickness with OCT may not be possible if this imaging does not show the interface between tumor and sclera, if the tumor is thicker than 2 mm or larger than 9 mm in LBD, or if it is located in the peripheral fundus [36,39].

### 4.5. Benefits of Omitting Ultrasonography from the Monitoring of Choroidal Nevi

Ultrasonography requires specialized equipment and an experienced operator and, as such, is often a limited resource in community settings. As a result, even when ocular oncology care is delivered via telemedicine platforms, patients are required to travel to centralized hubs where ultrasonography is available [6,7]. The omission of ultrasonography from the monitoring of choroidal nevi may aid in the implementation of tele-oncology programs, maximizing benefit to patients by decreasing both the time and cost associated with travel. Additionally, these programs would likely enhance the cost-utility of monitoring choroidal nevi [40]. Wide-field fundus photography, AF and OCT are commonly available in the community. While some degree of training is required to obtain high-quality fundus photographs and OCT, these skills can be easily taught to a technician and have a relatively short learning curve as, unlike US, minimal interpretation is required during image acquisition. Therefore, in an ideal scenario, a patient with a choroidal nevus could attend a local technician-led clinic where fundus photographs, AF and OCT would be obtained and reviewed remotely by an ophthalmologist. This is in contrast to many current protocols which require ultrasonography and, therefore, oblige patients to travel to a distant center for imaging.

In our study, only one case of progression (case 81) would have been missed had ultrasonography not been performed; however, the increase in thickness in this case was only 0.7 mm and may have represented measurement variability rather than true growth. As mentioned, this patient had clinical features strongly suggestive of melanoma and would therefore already have been under the care of an ocular oncologist.

Of course, there are situations in which ultrasonography is necessary to assess tumor thickness, including tumors that are too thick or too peripheral to be accurately assessed with OCT. Smaller lesions tend to grow disproportionately in basal dimensions compared to height, and thus are unlikely to increase ≥0.5 mm in thickness without also demonstrating an increase in basal dimensions; however, the monitoring of tumor thickness is more important for larger tumors (LBD > 6 mm), where LBD and thickness tend to increase in an approximately 1:1 ratio. Therefore, in larger tumors, occasionally an increase of ≥0.5 mm in thickness may be accompanied by only a subtle increase in basal dimensions.

### 4.6. Strengths and Weaknesses

The greatest strength of this study is the large number (96) of progressing melanocytic choroidal tumors. As our study included only tumors which were eventually treated, only three cases did not show any progression. This is not a weakness as the current study was not intended to investigate how many choroidal nevi show progression. A large retrospective study of nearly 4000 choroidal nevi showed malignant growth in only 90 cases [27].

However, as Singh et al. have shown, it is rare for choroidal tumors with a MOLES score of 0/1 to grow and, as such, a study assessing this subset of patients would have to be incredibly large in order to have enough cases of progression to draw any meaningful conclusions. It is also important to note that the MOLES scoring system is not intended to be used by ocular oncologists to select patients for treatment, but rather to guide non-specialists in optimizing monitoring and referral decisions. We reported tumor size only to describe the cohort. It was not our objective to compare devices (i.e., OCT versus US) with respect to their ability to measure tumor size within different ranges. Rather, OCT was used to evaluate the presence of sub-retinal fluid (S = 1). As such, we are unable to provide recommendations regarding the measurement of tumor thickness with OCT.

It is possible that tumors that progressed were excluded from this study because they were not treated. Such cases are likely to be very rare because all patients were assessed by experienced ocular oncologists in a tertiary referral center and regularly underwent assessment with serial color photographs, AF, OCT and ultrasonography.

### 4.7. Further Research

As all the imaging of cases included in this study was reviewed by authors with sub-specialty training in ocular oncology, further research is required to determine the ability of non-specialists to detect lateral extension and orange pigment. Additionally, all fundus photographs were obtained using wide-field fundus photography (Optos California). As discussed above, this platform combines green (532 nm) and red (635 nm) channels to create a pseudo-color image. Further research is required to determine if our results can be extrapolated to cases in which true color photography is employed. Finally, as only three tumors in this study did not progress, further study on a larger cohort of stable tumors is required to determine the specificity of assessment with color photography alone and in combination with AF and OCT.

## 5. Conclusions

In summary, the large majority of melanocytic choroidal tumors can be safely monitored without the routine need for ultrasonography. This appears to be particularly true for suspicious nevi, categorized as MOLES 1 or 2, in which progression could be detected using color photography alone in 100% of cases. Overall, the sensitivity for detecting progression with color photographs alone was 97% (95% CI 93–100%) and increased to 99% (95% CI 97–100%) by including autofluorescence and OCT. The findings of this study are particularly relevant in the current climate, where virtual care models are being adopted rapidly because of the COVID-19 pandemic.

## Figures and Tables

**Figure 1 cancers-12-01856-f001:**
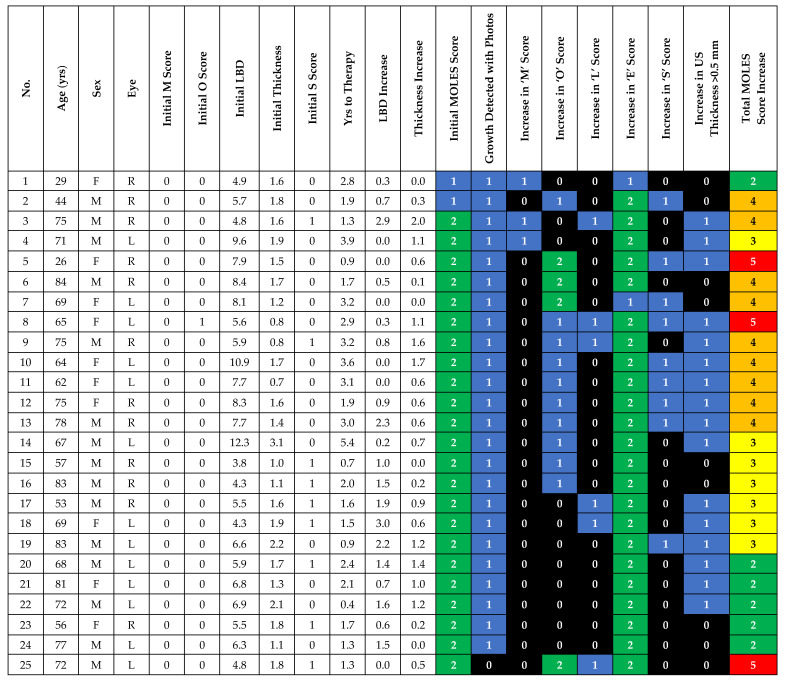
Demographic, baseline and progression details of MOLES 1/2 cases.

**Figure 2 cancers-12-01856-f002:**
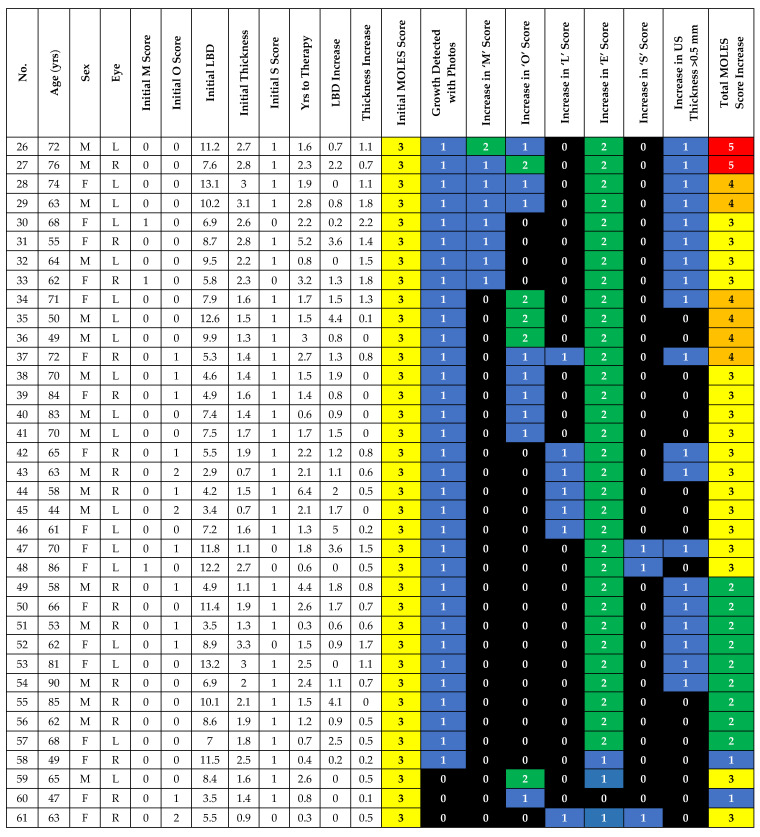
Demographic, baseline and progression details of MOLES 3 cases.

**Figure 3 cancers-12-01856-f003:**
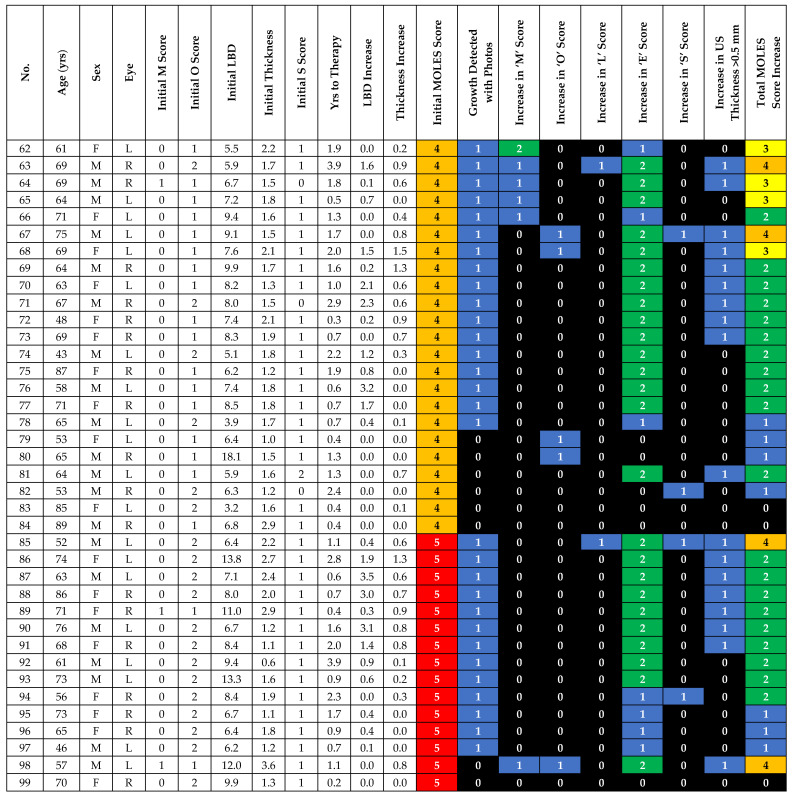
Demographic, baseline and progression details of MOLES 4/5 cases.

**Figure 4 cancers-12-01856-f004:**
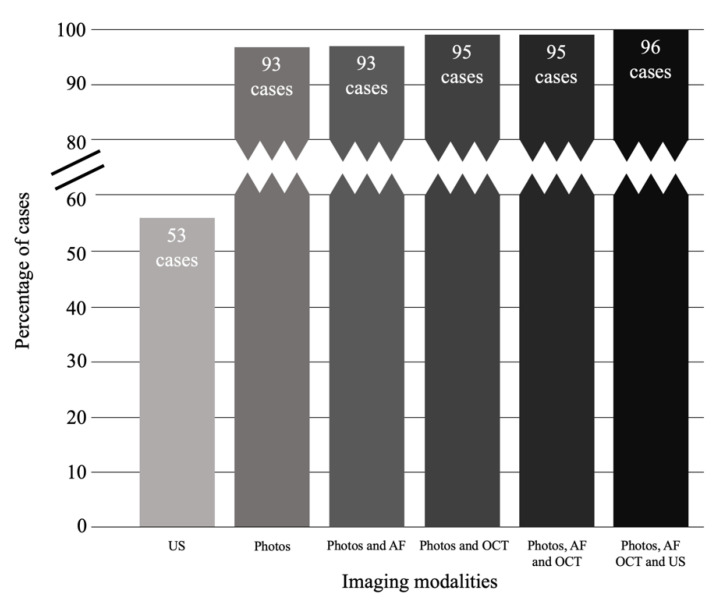
Progression of choroidal melanocytic tumors detected by various combinations of imaging modalities. Note: criteria for detection of progression on photos includes growth in basal dimensions and/or development of clumps of orange pigment (O = 2).

**Table 1 cancers-12-01856-t001:** MOLES scoring criteria.

Risk Factor	Severity	Score
**Mushroom Shape**	Absent	0
	Unsure/Early growth through RPE	1
	Present	2
**Orange Pigment**	Absent	0
	Unsure/Trace (i.e., Dusting)	1
	Confluent clumps	2
**Large Size ***	Thickness & Diameter	
	Thickness < 1.0 mm (‘flat/minimal thickening’) and diameter < 3DD	0
	Thickness = 1.0–2.0 mm (‘subtle dome shape’) and/or diameter = 3–4DD	1
	Thickness > 2.0 mm (‘significant thickening’) and/or diameter > 4DD	2
**Enlargement**	None (or lesion not documented or mentioned to patient previously)	0
	Unsure (i.e. Poor image quality)	1
	Definite (confirmed with sequential imaging)	2
**Subretinal Fluid ****	Absent	0
	Trace (if minimal and detected only with OCT)	1
	Definite (if seen without OCT)	2
	**Total Score**	

DD = disc diameter (= 1.5 mm); * ignore thickness if this cannot be measured; ** assume SRF if unexplained visual loss; MOLES = Mushroom, Orange pigment, Large size, Enlargement, and Subretinal fluid; RPE = retinal pigment epithelium; SRF = sub-retinal fluid; OCT = optical coherence tomography.

**Table 2 cancers-12-01856-t002:** MOLES tumor categories and recommended management.

MOLES Score	Suggested Management
**0 = Common Nevus**	**Monitoring in community** with color photography every 1–2 years.
**1 = Low-Risk Nevus** **2 = High-Risk Nevus**	**Non-urgent referral** for specialist investigation comprising wide-field photography, autofluorescence imaging, optical coherence tomography and, in selected cases, ultrasonography. Subsequent surveillance to be undertaken at a specialist clinic or in the community according to risk of malignancy.
**3 = Probable Melanoma**	**Urgent referral** to ophthalmologist with urgent onward referral to ocular oncologist if suspicion of malignancy is confirmed.

**Table 3 cancers-12-01856-t003:** Patient and tumor features at initial and final date.

Patient Features	Sub-Category	*n* (%)	
Sex	Female	46 (46.5)	
	Male	53 (53.5)	
Age (Yrs)	<66	47 (47.5)	
	>65	52 (52.5)	
Eye	Left	52 (52.5)	
	Right	47 (47.5)	
Treatment	Laser	2 (2.0)	
	Plaque	80 (80.8)	
	Proton beam	17 (17.2)	
**Tumor Features**		**Initial** (**%**)	**Final** (**%**)
Mushroom Shape	Nil	93 (93.9)	80 (80.8)
	Incipient	6 (6.1)	13 (13.1)
	Definite	0 (0.0)	6 (6.1)
Orange Pigment	Nil	47 (47.5)	24 (24.2)
	Dusting	30 (30.3)	34 (34.3)
	Confluent	22 (22.2)	41 (41.4)
Largest Basal Tumor Diameter (mm)	<4.5	10 (10.1)	5 (5.1)
	4.5–6.0	20 (20.2)	10 (10.1)
	>6.0	69 (69.7)	84 (84.8)
Tumor Thickness (mm)	<1.0	0 (0.0)	0 (0.0)
	1.0–2.0	74 (74.7)	43 (43.4)
	>2.0	25 (25.3)	56 (56.6)
Tumor Size	Small	2 (2.0)	0 (0.0)
	Borderline	26 (26.3)	12 (12.1)
	Large	71 (71.7)	87 (87.9)
Enlargement	Nil	99 (100)	8 (8.1)
	Possible	-	14 (14.1)
	Definite	-	77 (77.8)
Subretinal Fluid	Nil	25 (25.3)	12 (12.1)
	Minimal	73 (73.7)	83 (83.8)
	Significant	1 (1.0)	4 (4.0)
LBD Increase	<0.6 mm	-	40 (40.4)
	>0.5 mm	-	59 (59.6)
Thickness Increase	<0.6 mm	-	46 (46.5)
	>0.5 mm	-	53 (53.5)
Duration of Monitoring (Yrs)	<1	28 (28.3)	
	1–2	37 (37.4)	
	>2	34 (34.3)	
MOLES Score	0	0 (0.0)	0 (0.0)
	1	2 (2.0)	0 (0.0)
	2	23 (23.2)	0 (0.0)
	3	36 (36.4)	2 (2.0)
	4	23 (23.2)	8 (8.1)
	5	15 (15.2)	24 (24.2)
	>5	-	65 (65.7)

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
