# Peer review of "Detecting Progression of Melanocytic Choroidal Tumors by Sequential Imaging: Is Ultrasonography Necessary?"

_cancers, 2020, doi:10.3390/cancers12071856_

Round 1
Reviewer 1 Report
General
In the manuscript “Detecting progression of melanocytic choroidal tumors by sequential imaging: Is ultrasonography necessary?” the authors revealed the possibility of detect progression of choroidal melanocytic tumors with multi-modal imaging systems. They considered imaging with color photography, autofluorescence (AF), and optical coherence tomography (OCT). The analysis was performed on a representative group of patients.
The made analysis may find an interest and the study looks well-done, therefore it can be considered for publication in Cancers, with some necessary improvements, as pointed out below.
Specific
Line 23 - the MOLES, the acronym should be clarified in the abstract to better understanding.
Line 84 – FAF? shouldn’t it be AF?
Table 1a – RPE, SRF, no shortcut explanations found in the text.
Line 104 – LBD, no shortcut explanations found
Figure 1 and Figure 2 - some of the headings in the table headers are partially cut off.
Line 204 – EDI-OCT – EDI must be clarified.
The article is based on the examination of image information, unfortunately, no reference to the database in which the data is stored has been provided.
The introduction lacks references to the basic literature of the techniques discussed. For example, OCT is currently one of the key imaging techniques in ophthalmology. In addition, it is a component of multimodal systems. It would be useful if the techniques were initially discussed. Here are some example references:
Wollweber, M.; Roth, B. Raman Sensing and Its Multimodal Combination with Optoacoustics and OCT for Applications in the Life Sciences. Sensors 2019, 19, 2387.
Drexler, M. Liu, A. Kumar, T. Kamali, A. Unterhuber, R. A. Leitgeb, “Optical coherence tomography today: speed, contrast, and multimodality,” J. Biomed. Opt. 19, 071412 (2014)
There is no comparison between these techniques: imaging area, scanning depth, vertical and horizontal resolution, invasiveness, etc. Please provide also the device characteristics - manufacturer's name, NIR wavelength, etc.
There is no table regarding the general results of the study, which summarizes the range of tumor sizes that can be identified using US, AF, OCT, and color photographs.
The authors forecast the use of multimodal vision systems in telemedicine but do not specify how it can be used. Does the only US require an experienced operator? What about OCT? Please clarify it. The conclusions must be extended in this respect.
Author Response
Reviewer comment: In the manuscript “Detecting progression of melanocytic choroidal tumors by sequential imaging: Is ultrasonography necessary?” the authors revealed the possibility of detect progression of choroidal melanocytic tumors with multi-modal imaging systems. They considered imaging with color photography, autofluorescence (AF), and optical coherence tomography (OCT). The analysis was performed on a representative group of patients. The made analysis may find an interest and the study looks well-done, therefore it can be considered for publication in Cancers, with some necessary improvements, as pointed out below.
Response: Thank you for taking the time to review our manuscript and for providing us the opportunity to improve our work based on the comments below.
Reviewer comment: Line 23 - the MOLES, the acronym should be clarified in the abstract to better understanding.
Response: Thank you for this suggestion. We have modified this sentence to provide more clarity.
New text: MOLES scores (based on Mushroom shape, Orange pigment, Large size, Enlargement and Sub-retinal fluid) were assigned to all tumors at earliest date and date of treatment.
Reviewer comment: Line 84 – FAF? shouldn’t it be AF?
Response: Thank you for catching this inconsistence. We have changed the abbreviation to AF as it is throughout the rest of the paper.
Reviewer comment: Table 1a – RPE, SRF, no shortcut explanations found in the text.
Response: Thank you for identifying this omission. We have now added an explanation for both of these abbreviations to the footnote of Table 1a.
Reviewer comment: Line 104 – LBD, no shortcut explanations found
Response: Thank you for identifying this. We have now provided an explanation for LBD at its first use in the text.
Reviewer comment: Figure 1 and Figure 2 - some of the headings in the table headers are partially cut off.
Response: We have checked the figures to ensure the headings are correctly displayed.
Reviewer comment: Line 204 – EDI-OCT – EDI must be clarified.
Response: Thank you for highlighting this. An explanation for EDI-OCT is now provided.
New text: With small, posterior tumors, enhanced depth imaging OCT (EDI-OCT) may be a reasonable alternative in community practices.
Reviewer comment: The article is based on the examination of image information, unfortunately, no reference to the database in which the data is stored has been provided.
Response: Cases were identified by reviewing records of all patients undergoing treatment for choroidal melanoma at Moorfields Eye Hospital between January 2016 to March 2020. All images are stored on the Moorfields electronic patient health record under their respective imaging type. Images were reviewed directly from this system – they were not downloaded/exported to be stored in a separate database.
Reviewer comment: The introduction lacks references to the basic literature of the techniques discussed. For example, OCT is currently one of the key imaging techniques in ophthalmology. In addition, it is a component of multimodal systems. It would be useful if the techniques were initially discussed. Here are some example references:
Wollweber, M.; Roth, B. Raman Sensing and Its Multimodal Combination with Optoacoustics and OCT for Applications in the Life Sciences. Sensors 2019, 19, 2387.
Drexler, M. Liu, A. Kumar, T. Kamali, A. Unterhuber, R. A. Leitgeb, “Optical coherence tomography today: speed, contrast, and multimodality,” J. Biomed. Opt. 19, 071412 (2014)
Response: Thank you for highlighting this. We have provided a more thorough discussion on the basic literature of OCT and its advances over time and have cited both of these references. We have also added a sentence and reference regarding wide field imaging and autofluorescence.
New text: Over the past two decades, tremendous advances in both the resolution and speed of optical coherence tomography (OCT) imaging coupled with its non-invasive nature have led to its widespread adoption.20,21 Similarly, wide-field imaging modalities, including fundus autofluorescence, are being used increasingly to aide in the identification of lipofuscin.22 When used in combination with various other multi-modal imaging techniques, a great deal of information can be gathered. In the case of choroidal nevi, monitoring protocols often take advantage of this synergistic multi-modal approach by incorporating fundus photography, autofluorescence (AF), OCT and ultrasonography (US).
Reviewer comment: There is no comparison between these techniques: imaging area, scanning depth, vertical and horizontal resolution, invasiveness, etc. Please provide also the device characteristics - manufacturer's name, NIR wavelength, etc.
Response: Thank you for this helpful comment. We have elaborated further on all of the points you have brought up.
New text: Ocular imaging consisted of fundus photography and AF (Optos California [Optos plc, Dunfermline, Scotland]), OCT (Heidelberg Spectralis [Heidelberg Engineering GmbH, Heidelberg, Germany]) and US (Acuson Sequoia [Siemens, Healthineers, Frimley, Camberly, United Kingdom]. With respect to device specifics, the Optomap® pseudo-colour image was produced by combining the green (532nm) and red (635nm) channels whereas the autofluorescence image was generated only using the green. The Heidelberg Spectralis OCT used a wavelength of 870nm and the US was performed using a 15L8 transducer probe. These techniques complement one another as they vary in imaging area, scanning depth and invasiveness. For instance, wide field fundus photography and AF capture 200ï‚° of the fundus but are unable to provide detailed topographic information. Heidelberg OCT images a 30ï‚° field, but has the ability to provide vertical resolution far exceeding that of any other imaging modality. Finally, US has the greatest vertical scanning depth and is particularly useful for imaging lesions that are too thick for OCT (>2-3 mm); however, it is also the most invasive of the imaging modalities, requiring the probe to contact the eye.
Reviewer comment: There is no table regarding the general results of the study, which summarizes the range of tumor sizes that can be identified using US, AF, OCT, and color photographs.
Response: Thank you for your comment. We reported tumour size only to describe the cohort. It was not our objective to compare devices with respect to their ability to measure tumour size within difference ranges. Such comparisons would require several studies to adequately compare several devices with each other. Furthermore, these studies would have redundant as they have been done previously by other authors. For example, Sanket et al compared US and OCT with regards to thickness measurements. In any case, we did not find basal diameter measurement to be as sensitive as assessment of the relationship between tumour margins and adjacent landmarks. Also, size was only one indicator of tumour progression. A key message of our paper is that the development or increase in orange pigment and/or subretinal fluid were sensitive indicators of progression and these were demonstrable with colour photography and OCT. In fact, these indicated tumour progression much more often than ultrasound measurements of tumour thickness.
We used OCT to evaluate the presence of subretinal fluid and not to measure tumour thickness. There are two reasons for this. First of all, our objective was to determine whether monitoring is possible without US, because it is for ultrasonography that patients are usually referred to distant centres, not OCT, which is widely available. Second, OCT often does not allow accurate measurement of tumour thickness because it does not penetrate deep enough to demonstrate the interface between tumour and sclera, especially if the tumour is pigmented, because of the way that melanin absorbs light. As we did not use OCT to measure tumour size but rather, only to evaluate for the presence of absence of sub-retinal fluid, the results of this study are not specifically size dependent. We have endeavoured to clarify this further by adding a statement to our limitations section, indicating that we did not use OCT to measure tumour height and therefore, are unable to make recommendations for its use in this instance. Summary data of the patients included in this cohort is presented in table 2. In addition, the size data for each patient included in this study, along with their corresponding MOLES score, is illustrated in the figures.
New text: We reported tumour size only to describe the cohort. It was not our objective to compare devices (ie. OCT versus US) with respect to their ability to measure tumour size within difference ranges. Rather, OCT was used to evaluate the presence of sub-retinal fluid (S=1). As such, we are unable to provide recommendations regarding measurement of tumor elevation with OCT.
Reviewer comment: The authors forecast the use of multimodal vision systems in telemedicine but do not specify how it can be used. Does the only US require an experienced operator? What about OCT? Please clarify it. The conclusions must be extended in this respect.
Response: Thank you for this opportunity to further clarify our statements. We have provided additional text in the conclusion section to address this comment. While some degree of training is required to perform fundus photography and OCT, these techniques are far simpler and have a much shorter learning curve. Hence, many technicians are very capable of obtaining excellent images with minimal experience. As a result, these images could be obtained in the community and reviewed remotely by an ophthalmologist. Ultrasound is a current limitation to implementation of tele-oncology assessment because (i) the equipment is rarely available in the community, especially high-quality equipment providing the required resolution, and (ii) the skill required to obtain high quality images is far greater, because of the need to adjust gain appropriately, measure the thickest part of the tumour, avoid oblique scans and to gently hold the probe against the cornea without causing pain or damage. Hence, many patients often have to travel long distances to sub-specialty clinics primarily for ultrasonography. Omitting ultrasound would allow many patients to have their images acquired locally and reviewed by an ophthalmologist remotely. It is also our experience that many general ophthalmologists unnecessarily refer patients repeatedly for ultrasonography in their own department thereby overburdening this service when patients with small melanocytic choroidal tumours can be monitored perfectly adequately without these scans, as shown in our study. This insight has already improved the flow of patients at Oxford Eye Hospital, where one of the authors practises part time.
New text: While some degree of training is required to obtain high-quality fundus photographs and OCT, these skills can be easily taught to a technician and have a relatively short learning curve as, unlike US, minimal interpretation is required during image acquisition. Therefore, in an ideal scenario, a patient with a choroidal nevus could attend a local technician-led clinic where fundus photographs, AF and OCT would be obtained and reviewed remotely by an ophthalmologist. This is in contrast to many current protocols which require ultrasonography and therefore, require patients to travel to a distant center for imaging.

Reviewer 2 Report
This is a well-conducted study and the manuscript is well-written and a delight to read. I hope to improve the manuscript further with my few minor comments.
Table 2 - Difficult to see the limits of the sub-categories within each category. Perhaps the authors can improve the table with lines between each category.
It would improve the interpretation of the data if the authors can calculate sensitivity and specificity of progression detection without ultrasound but with AF&OCT (index test 1) or only fundus photos (index test 2) in comparison to the multimodal standard (reference test). 95% confidence intervals for these diagnostic test measures are important to include.
To avoid any confusion, the authors should discuss that OPTOS based fundus images (which are used in this study) are not true photographies, but are pseudocolor constructions of scanning laser ophthalmoscopy based images. This makes the results difficult to extrapolate to centers where true photographies may be used.
Author Response
REVIEWER 2:
Reviewer comment: This is a well-conducted study and the manuscript is well-written and a delight to read. I hope to improve the manuscript further with my few minor comments.
Response: Thank you very much for your kind comments and for taking the time to review our manuscript.
Reviewer comment: Table 2 - Difficult to see the limits of the sub-categories within each category. Perhaps the authors can improve the table with lines between each category.
Response: Thank you for this helpful comment. We have added lines between each category to make the table easier to read.
Reviewer comment: It would improve the interpretation of the data if the authors can calculate sensitivity and specificity of progression detection without ultrasound but with AF&OCT (index test 1) or only fundus photos (index test 2) in comparison to the multimodal standard (reference test). 95% confidence intervals for these diagnostic test measures are important to include.
Response: Thank you for this insightful suggestion. We have calculated sensitivity for photography alone vs in combination with AF and OCT and have added these results along with their corresponding 95% confidence intervals to several areas throughout the manuscript. As only three patients in this study did not demonstrate progression, we were unable to comment with respect to specificity. Although a ‘false positive’ (i.e., detection of progression on photos/AF/OCT) was not demonstrated in any of these cases, the fact that there were only three such instances makes us hesitant to cite a specificity of 100% as this would likely be an overestimation. We have added text to the section of our manuscript addressing further research to indicate that a study on a larger proportion of stable lesions would be optimal to determine the specificity of the proposed multi-modal imaging excluding ultrasonography.
New text: Therefore, in this study, the overall sensitivity for detecting progression with color photography alone was 97% (95% CI: 93 – 100%), improving to 99% (95% CI 97 – 100%) with autofluorescence and OCT.
New text: Overall, the sensitivity for detecting progression with color photographs alone was 97% (95% CI 93 – 100%) and increased to 99% (95% CI 97 – 100%) by including autofluorescence and OCT.
New text: Finally, as only three tumors in this study did not progress, further study on a larger cohort of stable tumors is required to determine the specificity of assessment with color photography alone and in combination with AF and OCT.
Reviewer comment: To avoid any confusion, the authors should discuss that OPTOS based fundus images (which are used in this study) are not true photographies, but are pseudocolor constructions of scanning laser ophthalmoscopy based images. This makes the results difficult to extrapolate to centers where true photographies may be used.
Response: Thank you for highlighting this important point. We have expanded our discussion in the methods section to highlight the pseudo-color nature of optos photographs. We have also added a comment near the end of the manuscript to highlight the need for additional research in order to determine if the results from this study can be applied to instances where true color photographs are employed.
New text: As discussed above, this platform combines green (532nm) and red (635nm) channels to create a pseudo-color image. Further research is required to determine if our results can be extrapolated to cases in which true color photography is employed.

Reviewer 3 Report
The authors want to determine if ultrasonography is necessary to detect progression of choroidal melanocytic tumors undergoing sequential multi-modal imaging with color photography, autofluorescence (AF) and optical coherence tomography (OCT).
There are some questions should also be clarified.
- Who assess the initial and final MOLES scores?
- When assess the final MOLES scores MOLES score
- In 1/36 (2.85% of these tumors 125 (case 61), progression was demonstrated only by the appearance of trace sub-retinal fluid (S=1), visible on “OCT” (1/36; 2.8%) together with an increase in thickness on “US” of 0.5 mm. à Why the authors conclude that 127 progression would have been detected without ultrasonography in all 36 (100%) tumors (?) and without AF or OCT in 35 (97%). Did it mean OCT can replace US?
- In patients with MOLES 4 or 5, progression was detected by development of trace sub-retinal fluid detected on “OCT” in 1/38 (2.6%) or an isolated increase in thickness of 0.7 mm detected on US in 1/38 (2.6%). à Why the authors conclude this change was detected without ultrasonography in 135 34/35 (97.1%) cases and on fundus photography alone, without the assistance of AF or “OCT” in 33/35 136 (94%) cases?
Author Response
REVIEWER 3:
Reviewer comment: The authors want to determine if ultrasonography is necessary to detect progression of choroidal melanocytic tumors undergoing sequential multi-modal imaging with color photography, autofluorescence (AF) and optical coherence tomography (OCT). There are some questions should also be clarified. Who assess the initial and final MOLES scores?
Response: The initial and final MOLES scores were assessed by either KR or RO. Please see the new text below the next comment which addresses both points.
Reviewer comment: When assess the final MOLES scores MOLES score
Response: Thank you for allowing us to provide further clarification. We have added additional information to the methods section.
New text: Scoring of earliest images available and last images preceding treatment was performed by either KR or RO, both of whom have sub-specialty training in ocular oncology.
Reviewer comment: In 1/36 (2.85% of these tumors 125 (case 61), progression was demonstrated only by the appearance of trace sub-retinal fluid (S=1), visible on “OCT” (1/36; 2.8%) together with an increase in thickness on “US” of 0.5 mm. à Why the authors conclude that 127 progression would have been detected without ultrasonography in all 36 (100%) tumors (?) and without AF or OCT in 35 (97%). Did it mean OCT can replace US?
Response: Thank you for allowing us to clarify these conclusions. We have re-worded this paragraph to make the results more clear. In this group of MOLES-3 tumours, all but one (35/36) demonstrated progression on color photography alone. The final patient had progression detected on OCT (1/36). Therefore, in this group, progression would have been detected without the help of ultrasonography in all 36 cases. OCT did not replace US, because it measured a different indicator of progression than US (i.e., OCT assessed SRF and US assessed thickness). What OCT did was to make US unnecessary, because OCT was more sensitive at revealing progression than US, in other words detecting increasing SRF before US demonstrated increasing thickness in all except 1 case.
Revised text: A total of 36 cases had an initial MOLES score of 3 (36/99; 36.4%). In these cases, tumor progression was detected on color photography alone by (a) lateral tumor extension in 33/36 (91.7%) or (b) development of clumps of orange pigment (O=2) in 2/36 (5.6%). Therefore, progression would have been detected using color photography alone in 35/36 cases (97.2%; 95% CI: 92 – 100%). In the final remaining case (case 61), progression was demonstrated by the appearance of trace sub-retinal fluid (S=1), visible on OCT, together with an increase in thickness on US of 0.5 mm. As OCT allowed detection of tumour progression in the only case without lateral extension on color photography, progression would have been detected without ultrasonography in all 36 (100%) tumors. (Figure 2)
Reviewer comment: In patients with MOLES 4 or 5, progression was detected by development of trace sub-retinal fluid detected on “OCT” in 1/38 (2.6%) or an isolated increase in thickness of 0.7 mm detected on US in 1/38 (2.6%). à Why the authors conclude this change was detected without ultrasonography in 135 34/35 (97.1%) cases and on fundus photography alone, without the assistance of AF or “OCT” in 33/35 136 (94%) cases?
Response: Thank you for the opportunity to clarify our reasoning. We have re-worded this paragraph as below.
Revised text: There were 38 lesions categorized as MOLES 4 or 5. Three lesions did not show any signs of progression (cases 83, 84 and 99). In cases 83 and 99, choroidal melanoma was diagnosed at the first visit but the patients initially refused treatment and after a delay of 0.4 and 0.2 years, respectively, they agreed to undergo plaque brachytherapy. Case 84 underwent a work-up to rule out a systemic primary and following negative investigations, was treated with plaque brachytherapy after a delay of 0.4 years. In the remaining 35 tumors progression occurred and was detected on color photography by (a) lateral extension in 30/35 (85.7%) or (b) an increase in orange pigment to O=2 in 3/35 (8.6%). Therefore, color photography alone would have identified progression in 33/35 cases (94.3%; 95% CI, 87 – 100%). In one of the two remaining cases, progression was detected by the development of trace sub-retinal fluid detected on OCT in 1/35 (2.9%). Therefore, using a combination of color photography, AF and OCT, progression was detected without ultrasonography in 34/35 cases (97.1%). Only 1/35 (2.9%) tumor in this group (Case 81) demonstrated an increase in thickness (i.e., 0.7 mm), which was detected on US without visible change on color photography or OCT. (Figure 3)
